# Vitamin D Concentration and Motoric Cognitive Risk in Older Adults: Results from the Gait and Alzheimer Interactions Tracking (GAIT) Cohort

**DOI:** 10.3390/ijerph192013086

**Published:** 2022-10-12

**Authors:** Maxime Le Floch, Jennifer Gautier, Cédric Annweiler

**Affiliations:** 1Research Center on Autonomy and Longevity, UPRES EA 4638, Department of Geriatric Medicine, Angers University Hospital, University of Angers, F-49000 Angers, France; 2Health Faculty, School of Medicine, University of Angers, F-49000 Angers, France; 3Department of Medical Biophysics, Robarts Research Institute, Schulich School of Medicine and Dentistry, University of Western Ontario, London, ON N6A 3K7, Canada

**Keywords:** vitamin D deficiency, aged, slower gait, subjective memory complaint

## Abstract

Background. Motoric Cognitive Risk (MCR) syndrome, which combines subjective memory complaint (SMC) and slower gait speed, is a newly-described predementia stage. Based on the involvement of vitamin D in the biology of both gait and cognition, we hypothesized that nondemented individuals with MCR would exhibit hypovitaminosis D more often compared to Cognitively Healthy Individuals (CHI). The objective of this cross-sectional analysis was to determine whether hypovitaminosis D was associated with MCR. Methods. Participants without dementia from the GAIT (Gait and Alzheimer Interactions Tracking) cohort study were classified into MCR or Cognitively Healthy Individuals (CHI) groups. Hypovitaminosis D was defined as the lowest quartile of serum 25-hydroxyvitamin D (25OHD) concentration compared to the other three combined. Age, sex, body mass index (BMI), the Frontal Assessment Battery (FAB) score, the Mini-Mental Short Examination (MMSE) score, education level, use of psychoactive drugs, and the number of chronic diseases were used as covariates. Results. Among 244 nondemented and nonMCInonMCR participants from the GAIT cohort (mean age 71.4 ± 3.7 years, 40.6% women), 66 participants were classified as MCR (36.9%) and 178 as CHI (63.1%). The lowest quartile of 25OHD concentration was directly associated with MCR (unadjusted OR = 2.85, *p* = 0.003) even after adjustment for studied potential confounders (fully adjusted OR = 2.61, *p* = 0.025). The BMI (adjusted OR = 6.65, *p* < 0.001), MMSE score (adjusted OR = 0.74, *p* = 0.009), FAB score (adjusted OR = 0.51, *p* < 0.001), number of chronic diseases (adjusted OR = 1.29, *p* = 0.043) and use of psychoactive drugs (adjusted OR = 2.55, *p* = 0.044) were also associated with MCR. Conclusions. Hypovitaminosis D was associated with MCR in older community-dwellers without dementia.

## 1. Introduction

In the perspective of novel disease-modifying treatments, there is a need for detecting people with cognitive impairment at a pre-dementia stage. The Motoric Cognitive Risk syndrome (MCR syndrome) is a newly-described clinical approach of the interactions between cognitive and gait disorders [1,2]. Briefly, the MCR syndrome combines subjective memory complaint (SMC) and slower gait speed. MCR is a risk factor for cognitive decline and dementia [3,4] and for various geriatric syndromes including falls [5,6]. MCR is common in older populations with a prevalence of 9.7% in a pooled analysis of 26,802 older adults from 17 countries and an incidence of 65.2/1000 persons in four US (United States)-based cohorts [1,2].

Besides its classical function of bone metabolism regulation, vitamin D exhibits various biological targets mediated by the Vitamin D Receptor (VDR) [7]. Specific actions on target organs such as the muscles and the central nervous system, directly influencing motor and cognitive performances, have been previously reported [8,9]. As a consequence, hypovitaminosis D, which is highly frequent in older adults [10], has been repeatedly associated with outcomes such as slower gait speed [11], falls [10,12], SMC [13,14], and cognitive disorders [15].

Based on the involvement of vitamin D in the biology of both gait and cognition, we hypothesized that nondemented individuals with MCR would exhibit hypovitaminosis D more often than Cognitively Healthy Individuals (CHI). Better understanding this relationship appears crucial to further develop interventions testing the effect of vitamin D supplementation on MCR prognosis and outcomes. The aim of the present study was to determine whether hypovitaminosis D was associated with MCR in a cohort of older adults with SMC but no dementia.

## 2. Materials and Methods

### 2.1. Participants

A total of 912 individuals were recruited in the “Gait and Alzheimer Interactions Tracking” (GAIT) study, which is a large French cross-sectional study testing the neuroanatomical correlates of gait in older adults with SMC. The study design and assessments have been previously described in detail [16]. All eligible participants were referred to the memory clinic of Angers University Hospital, France, for an evaluation of SMC. The GAIT eligibility criteria were aged 60 years and older, being affiliated to a social security regime, SMC, an adequate understanding of French, able to walk without a walking aid for 15 m, Mini Mental State Examination (MMSE) score > 10, near visual acuity ≥ 2/10, and absence of severe depression. Exclusion criteria included acute medical illness in the past 3 months, extrapyramidal rigidity of the upper limbs, and neurological and psychiatric diseases other than cognitive disorder. Inclusion criteria in the present analysis were as follows: recruitment in Angers, France, no diagnosis of dementia, diagnosis of MCR or CHI, measure of serum 25-hydroxyvitamin D (25OHD) at the time of cognitive, and gait assessments. Those with Mild Cognitive Impairment (MCI) but no MCR were retained to define slower gait in our study but not in the between-groups analysis. The diagnoses of MCI and dementia were made during multidisciplinary meetings involving geriatricians, neurologists, and neuropsychologists of the memory clinic of Angers University Hospital and were based on a review of all available neuropsychological tests, physical examination findings, blood test results, and applying the consensual Diagnostic and Statistical Manual of Mental Disorders, fourth edition, (DSM-IV) criteria [17] and Winblad criteria [18].

### 2.2. Motoric Cognitive Risk Syndrome

The MCR diagnosis builds on MCI criteria [19] and is defined as the presence of SMC and slower gait in older individuals without dementia or motor disabilities [2,3]. As SMC was the reason for referral to the memory clinic of Angers University Hospital, France, all participants met this criterion. Gait speed was measured with GAITRite (Gold walkway, 972 cm long, active electronic surface area 792 × 610 cm, total 29,952 pressure sensors, scanning frequency 60 Hz; CIR System, Havertown, PA, United State of America. Slower gait was defined as gait speed 1 standard deviation below age- and sex-specific means in the GAIT cohort [20]. Cut-scores for defining slower gait speed in our cohort are reported in Table 1.

### 2.3. Vitamin D Status

Blood samples were collected from participants during the first examination. Serum 25OHD concentration, which accurately reflects the vitamin D status, was measured using radioimmunoassay (DiaSorin Inc., Stillwater, MN, USA) in nmol/L locally at the University Hospital of Angers, France. Serum 25OHD concentrations were classified into quartiles with an equal number of participants in each quartile: <42, [42; 57[, [57; 74[, ≥74 (to convert to ng/mL, divide by 2.496) [21]. Hypovitaminosis D was defined as the lowest quartile of serum 25OHD concentration compared to the other three quartiles combined.

### 2.4. Covariates

Age, body mass index (BMI), sex, education level, Mini-Mental States Evaluation (MMSE) score, Frontal Assessment Battery (FAB) score, number of chronic diseases, and use of psychoactive drugs (including benzodiazepines, antidepressants, hypnotics or neuroleptics) were included in this data analysis as potential confounders. BMI was calculated as weight (kg) divided by height^2^ (m^2^). Assessment of chronic diseases was based on self-reported and structured health status questionnaires. Chronic diseases were diseases lasting at least 3 months or running a course with minimal change, whatever their nature or site. The psychoactive drugs usually taken were determined from the primary care physician’s prescriptions and by questioning the patient, whatever the dosage schedule or route of administration and regardless of the date of commencement. Education level was self-reported with a structured standardized questionnaire. Participants who did graduate studies were considered to have a higher level of education compared with those who did not. Finally, global cognitive performance was assessed with the MMSE score, and executive functions with the FAB score [22,23].

### 2.5. Registrations

The study was conducted in accordance with the ethical standards set forth in the Helsinki Declaration (1983). The entire study protocol was approved by the Angers Ethical Review Committee (Comité de protection des personnes, CPP ouest II, Angers, France, n°2009-A00533-54). Written informed consent was obtained at enrollment (clinicaltrials.gov Identifier: NCT01315717).

### 2.6. Statistics

Participants’ characteristics were summarized using means and standard deviations (SD) or frequencies and percentages, as appropriate. First, participants were categorized into two groups: those with MCR and those defined as cognitively healthy individuals (CHI) i.e., nondemented, nonMCI and nonMCR. Between-group comparisons were performed using the chi-square test or the independent samples t test, as appropriate. Second, univariate and multiple (i.e., fully adjusted) logistic regressions were performed to assess the association between MCR (dependent variable) and hypovitaminosis D (independent variable). Nine covariates were used in multiple analyses; *p*-values < 0.05 were considered significant.

## 3. Results

A total of 912 participants were included in the GAIT cohort. Among them, 181 were not included in Angers Hospital, two were younger than 65 years, 132 had a diagnosis of dementia, and 208 unpreserved activities of daily living. Finally full data were available for 389 participants, whose gait tests were used to determine the cut-score for slower gait speed. Table 1 shows cut-scores to define MCR for each age and sex-groups.

In total, 244 participants with either MCR (*n* = 66) or CHI (*n* = 178) were included in the present analyses. The prevalence of MCR was 16.9% in the present cohort. As indicated in Table 2, the participants with MCR more often exhibited hypovitaminosis D than the CHI (*p* = 0.002). They also had a lower MMSE score (*p* < 0.001), FAB score (*p* < 0.001), and education level (*p* = 0.016), a higher BMI (*p* < 0.001), more chronic diseases (*p* < 0.001), and used more often psychoactive drugs (*p* = 0.004) than the others. There were no significant differences for the other characteristics.

Table 3 reports univariate and multiple logistic regressions between MCR, participants’ characteristics, and hypovitaminosis D. Hypovitaminosis D was directly associated with MCR (unadjusted OR = 2.85, *p* = 0.003) and even after adjustment for potential confounders (fully adjusted OR = 2.61, *p* = 0.025). Additionally, BMI (adjusted OR = 6.65, *p* < 0.001), MMSE score (adjusted OR = 0.74, *p* = 0.009), FAB score (adjusted OR = 0.51, *p* < 0.001), number of chronic diseases (adjusted OR = 1.29, *p* = 0.043), and the use of psychoactive drugs (adjusted OR = 2.55, *p* = 0.044) were also associated with MCR.

## 4. Discussion

The main finding of this study is that hypovitaminosis D lower than 42 nmol/L was associated with MCR syndrome in the studied sample of older nondemented community-dwellers with SMC after adjusting for all studied potential confounders. This result paves the way to conduct further interventions testing the effect of vitamin D supplementation on MCR prevention and treatment.

Our study provides the first evidence of an association between vitamin D and MCR syndrome in older adults without dementia. The biological pathways leading to MCR are not fully characterized. Since one component of MCR syndrome is SMC, no animal model can help elucidating its pathophysiology. Two main theories may explain the association between a lower circulating 25OHD concentration and MCR syndrome. First, it is possible that slower gait speed and cognitive disorders met in MCR may result in reduced sunlight exposure and dietary vitamin D intakes due to loss of functional abilities and may ultimately cause hypovitaminosis D [24,25]. However, the BMI of MCR here was higher on average than that of CHI. A scenario of reverse causation should thus be considered, with hypovitaminosis D resulting in MCR syndrome. Vitamin D is a biological determinant of motor abilities and gait speed [11]. Lower 25OHD concentrations were associated with biological parameters implicated in gait disorders such as poorer lower-extremity function, lower muscle strength, lower contraction speed, and slower nerve conduction [11]. Lower baseline 25OHD concentrations are associated with greater decline in gait speed in longitudinal studies [26]. In contrast, interventions have reported gait improvement following vitamin D supplies [27]. Such effect may be explained by the involvement of vitamin D in muscular health and function and also in higher-level motor control [28,29]. Vitamin D enters the cerebrospinal fluid through passive diffusion of the blood–brain barrier [30]. It exerts its actions through a specific receptor, the VDR, which is mostly expressed in key regions for cognition such as the hippocampus and hypothalamus [31]. In fact, vitamin D regulates the gene expression of several neurotrophins and neurotransmitters, as well as various oxidative and inflammatory changes in the brain [11,32]. This neurosteroid involvement of vitamin D appears important regarding the link with SMC [13]. This may explain why hypovitaminosis D has been previously associated with SMC, cognitive decline, dementia onset in longitudinal studies, and with brain atrophic changes and white matter abnormalities [14,33,34,35,36]. Thus, finding an association between vitamin D and MCR syndrome is unprecedented but not surprising.

The different gait cut-scores defining MCR syndrome in our cohort were higher than those in previous literature [20]. This may be explained by the fact that Parkinsonism and the history of stroke, two diseases well-known to negatively influence gait speed, were exclusion criteria in our cohort. Consistently, our cut-scores were similar to those of another cohort using similar exclusion criteria [5]. Similarly, the relatively high prevalence of MCR syndrome (16.9%) in our cohort of people with SMC was greater than that in the general population (estimated around 9.7%), but close to another cohort enrolling seniors with SMC [2].

To our knowledge, we provide here the first study on the association between vita-min D and MCR syndrome. However, the limitations of our study should be acknowledged. First, it took place in one single memory center and included only community-dwellers with SMC. Thus, the studied population may be not representative of the general population of older adults, as illustrated by the relatively high MCR prevalence. Second, although we were able to control for many characteristics likely to modify the association between vitamin D and MCR syndrome, residual confounders such as the concentrations of parathormone or calcium might be still present. Third, the cross-sectional design of our study does not allow any causal inference.

## 5. Conclusions

MCR is an important approach of the long predementia phase, with clinical and re-search implications. We were able to report a direct association between hypovitaminosis D and MCR syndrome in older adults without dementia, especially below 42 nmol/L. This result should be confirmed in larger longitudinal studies, preferentially on a variety of adult populations. Nevertheless, it provides a scientific basis for conducting clinical trials to test the efficacy of vitamin D supplementation to prevent or improve the prognosis of MCR in older patients with initial serum 25OHD < 42 nmol/L.

## Figures and Tables

**Table 1 ijerph-19-13086-t001:** Gait speed * cut-scores for defining slower gait speed for the Motoric Cognitive Risk (MCR) syndrome.

Age Group (y)	*n*	Men	*n*	Women
[65–69]	96	98.82	62	87.25
[70–74]	87	90.50	55	86.12
[75–79]	47	90.70	24	82.01
≥80	9	72.63	9	75.87

y: years; n: number; * gait speed (cm/s).

**Table 2 ijerph-19-13086-t002:** Characteristics and comparison according to the MCR syndrome (*n* = 244).

	Total Cohort (*n* = 244)	CHI (*n* = 178)	MCR (*n* = 66)	*p*-Value ***
Age (year), mean ± SD	71.4 ± 3.7	71.2 ± 3.5	71.8 ± 4.2	0.574
Female sex, *n* (%)	99 (40.6)	74 (41.6)	25 (37.9)	0.602
Body mass index, n (%)				**<0.001**
<21 kg/m^2^	17 (7.0)	14 (7.9)	3 (4.6)	
[21–30[ kg/m^2^	190 (77.9)	149 (83.7)	41 (62.1)	
≥30 kg/m^2^	37 (15.2)	15 (8.4)	22 (33.3)	
High education level ^†^, n (%)	145 (59.4)	114 (64.0)	31 (47.0)	**0.016**
MMSE score (/30), mean ± SD	28.0 ± 1.7	28.4 ± 1.4	27.1 ± 1.9	**<0.001**
FAB score (/18), mean ± SD	16.3 ± 1.6	16.7 ± 1.2	15.1 ± 2.1	**<0.001**
Number of chronic diseases, mean ± SD	2.0 ± 1.5	1.8 ± 1.4	2.6 ± 1.7	**<0.001**
Use of psychoactive drugs ^‡^, n (%)	42 (17.2)	23 (12.9)	19 (28.8)	**0.004**
Hypovitaminosis D ^||^, n (%)	59 (24.2)	34 (19.1)	25 (37.9)	**0.002**

25OHD: 25-hydroxyvitamin D; CHI: cognitively healthy individuals; FAB: Frontal Assessment Battery; MCR: motoric cognitive risk; MMSE: Mini-Mental State Examination; SD: standard deviation; * Chi^2^ test or Fisher’s exact test was used for qualitative variables, as appropriate; Student *t* test or Mann–Whitney Wilcoxon test for quantitative variables, as appropriate; ^†^: graduate studies; ^‡^: antidepressants, neuroleptics, benzodiazepines, or hypnotics; ^||^: serum 25OHD concentration < 42 nmol/L; *p*-value significant (i.e., *p* < 0.05) indicated in bold.

**Table 3 ijerph-19-13086-t003:** Univariate and multiple logistic regressions showing associations between Motoric Cognitive Risk syndrome (dependent variable) and participants’ characteristics including hypovitaminosis D (independent variables) (*n* = 244).

	Unadjusted Model		Fully Adjusted Model	
	OR [95% CI]	*p*-Value	OR [95% CI]	*p*-Value
Hypovitaminosis D *	2.58 [1.39; 4.81]	**0.003**	2.61 [1.13; 6.06]	**0.025**
Female sex	0.67 [0.48; 1.53]	0.602	1.06 [0.49; 2.32]	0.881
Age	1.04 [0.97; 1.12]	0.283		0.201
Body mass index		**<0.001**		**<0.001**
<21 kg/m^2^	0.78 [0.21; 2.84]		1.17 [0.25; 5.39]	
[21–30] kg/m^2^	1		1	
≥30 kg/m^2^	5.33 [2.54; 11.19]		6.65 [2.62; 16.87]	
High education level ^†^	0.50 [0.28; 0.88]	**0.017**	0.77 [0.36; 1.63]	0.490
MMSE score	0.63 [0.52; 0.76]	**<0.001**	0.74 [0.59; 0.93]	**0.009**
FAB score	0.52 [0.42; 0.65]	**<0.001**	0.51 [0.39; 0.68]	**<0.001**
Number of chronic diseases	1.39 [1.15; 1.67]	**<0.001**	1.29 [1.01; 1.65]	**0.043**
Use psychoactive drugs ^‡^	2.72 [1.37; 5.43]	**0.004**	2.55 [1.03; 6.36]	**0.044**

25OHD: 25-hydroxyvitamin D; CI: confidence interval; FAB: Frontal Assessment Battery; MCR: motoric cognitive risk; MMSE: Mini-Mental State Examination; OR: odds ratio; SD: standard deviation; *: serum 25OHD concentration < 42 nmol/L; ^†^: graduate studies; ^‡^: Antidepressants, neuroleptics, benzodiazepines, or hypnotics; odds ratio significant (i.e., *p* < 0.05) indicated in bold.

## Data Availability

Patient level data are freely available from the corresponding author at CeAnnweiler@chu-angers.fr. There is no personal identification risk within these anonymized raw data, which are available after notification and authorization of the competent authorities.

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
