# Peer review of "Vitamin D Concentration and Motoric Cognitive Risk in Older Adults: Results from the Gait and Alzheimer Interactions Tracking (GAIT) Cohort"

_ijerph, 2022, doi:10.3390/ijerph192013086_

Round 1

Reviewer 1 Report

The manuscript entitled "Vitamin D concentration and motoric cognitive risk in older adults: results from the gait and alzheimer interactions tracking (GAIT) cohort" provides a findings on the association of vitamin D deficiency with MCR.

I present my comments and suggestions for changes in relation to the following parts of the article.

(Line 40, 41) Please use the same terms.

- (Line 40) MCR Syndrome

- (Line 41) MCR syndrome

(Line 70) (... MMSE score ...) Please include the full terms before using an abbreviation.

(Line 81) (... DSM-IV criteria ...) Please include the full terms before using an abbreviation.

(Line 90) ("Participants were divided into 8 age ...") In Table 1, there are 4 groups with an age difference of 4 years. I don't know what this sentence means.

(Line 97) (... participants in each quartile: <42, [42; 57[, [57; 74[, ≥ 74 ...) I don't know what this sentence means.

(Line 105) (... divided by height2 (m2).) Please check the unit notation.

(Line 130) These sentences seems to have grammatical problems. I think you need to correct the part where there is a problem with the grammar.

(Line 144-145) Is there a reason to use "Mean – 1 SD", in Table 1? Please explain this.

(Line 147) In Table 2, there are several places that need to be modified.

- (<21 kg/m2, 17(7.0), 25(37.9)) Please check the space.

- ([21-30[)Is it correct to display like this? Is [21-30] spelled incorrectly as [21-30[?

Reviewer 2 Report

Very well presented study, centered on the determination of the existence of any association between Vitamin D concentration and Motoric Cognitive Risk in older adults.

I could only raise a concern, centered on the fact that the aforementioned results are derived from the Gait and Alzheimer Interactions Tracking (GAIT) Cohort. The validity of the study could be enhanced if the sample of participants would be extracted from a community that included a more generalized patient population.

The main  question addressed by the research was to elucidate hypovitaminosis D was associated with Motoric Cognitive Risk (MCR) syndrome. The topic is original in the field as, to the best of my knowledge, there are no known relevant studies centered on the detection of any association between vitamin D and MCR syndrome in older adults without dementia. This study adds a novel finding to the subject area compared with other published material, as no other similar studies are published.

I strongly consider that the conclusions are consistent with the evidence and arguments presented and they adequately address the main question posed. It should be mentioned that this is a preliminary study. I state that the references are appropriate and in the correct style.  I do not feel that any additional comments on the tables and figures are needed.

Round 2

Reviewer 1 Report

Overall, the manuscript is very well written and organized.

Thank you for your effort.